# GROUP-CONNECTED MULTILAYER PERCEPTRON NETWORKS

## ABSTRACT

Despite the success of deep learning in domains such as image, voice, and graphs, there has been little progress in deep representation learning for domains without a known structure between features. For instance, a tabular dataset of different demographic and clinical factors where the feature interactions are not given as a prior. In this paper, we propose Group-Connected Multilayer Perceptron (GMLP) networks to enable deep representation learning in these domains. GMLP is based on the idea of learning expressive feature combinations (groups) and exploiting them to reduce the network complexity by defining local group-wise operations. During the training phase, GMLP learns a sparse feature grouping matrix using temperature annealing softmax with an added entropy loss term to encourage the sparsity. Furthermore, an architecture is suggested which resembles binary trees, where group-wise operations are followed by pooling operations to combine information; reducing the number of groups as the network grows in depth. To evaluate the proposed method, we conducted experiments on five different real-world datasets covering various application areas. Additionally, we provide visualizations on MNIST and synthesized data. According to the results, GMLP is able to successfully learn and exploit expressive feature combinations and achieve state-of-the-art classification performance on different datasets.

## 1    INTRODUCTION

Deep neural networks have been quite successful across various machine learning tasks. However, this advancement has been mostly limited to certain domains. For example in image and voice data, one can leverage domain properties such as location invariance, scale invariance, coherence, etc. via using convolutional layers (Goodfellow et al., 2016). Alternatively, for graph data, graph convolutional networks were suggested to leverage adjacency patterns present in datasets structured as a graph (Kipf & Welling, 2016; Xu et al., 2019).

However, there has been little progress in learning deep representations for datasets that do not follow a particular known structure in the feature domain. Take for instance the case of a simple tabular dataset for disease diagnosis. Such a dataset may consist of features from different categories such as demographics (e.g., age, gender, income, etc.), examinations (e.g., blood pressure, lab results, etc.), and other clinical conditions. In this scenario, the lack of any known structure between features to be used as a prior would lead to the use of a fully-connected multilayer perceptron network (MLP). Nonetheless, it has been known in the literature that MLP architectures, due to their huge complexity, do not usually admit efficient training and generalization for networks of more than a few layers.

In this paper, we propose Group-Connected Multiplayer Perceptron (GMLP) networks. The main idea behind GMLP is to learn and leverage expressive feature subsets, henceforth referred to as *feature groups*. A feature group is defined as a subset of features that provides a meaningful representation or high-level concept that would help the downstream task. For instance, in the disease diagnosis example, the combination of a certain blood factor and age might be the indicator of a higher level clinical condition which would help the final classification task. Furthermore, GMLP leverages feature groups limiting network connections to local group-wise connections and builds a feature hierarchy via merging groups as the network grows in depth. GMLP can be seen as an architecture that learns expressive feature combinations and leverages them via group-wise operations.

The main contributions of this paper are as follows: ($i$) proposing a method for end-to-end learning of expressive feature combinations, ($ii$) suggesting a network architecture to utilize feature groups and

local connections to build deep representations, $(iii)$ conducting extensive experiments demonstrating the effectiveness of GMLP as well as visualizations and ablation studies for better understanding of the suggested architecture.

We evaluated the proposed method on five different real-world datasets in various application domains and demonstrated the effectiveness of GMLP compared to state-of-the-art methods in the literature. Furthermore, we conducted ablation studies and comparisons to study different architectural and training factors as well as visualizations on MNIST and synthesized data. To help to reproduce the results and encouraging future studies on group-connected architectures, we made the source code related to this paper available online [1].

## 2 RELATED WORK

Fully-connected MLPs are the most widely-used neural models for datasets in which no prior assumption is made on the relationship between features. However, due to the huge complexity of fully-connected layers, MLPs are prone to overfitting resulting in shallow architectures limited to a few layers in depth (Goodfellow et al., 2016). Various techniques have been suggested to improve training these models which include regularization techniques such as L-1/L-2 regularization, dropout, etc. and normalization techniques such as layer normalization, weigh normalization, batch normalization, etc.(Srivastava et al., 2014; Ba et al., 2016; Salimans & Kingma, 2016; Ioffe & Szegedy, 2015). For instance, self-normalizing neural networks (SNNs) have been recently suggested as state of the art normalization methods that prevent vanishing or exploding gradients which help training feed-forward networks with higher depths (Klambauer et al., 2017).

From the architectural perspective, there has been great attention toward networks consisting of sparse connections between layers rather than having dense fully-connected layers (Dey et al., 2018). Sparse connected neural networks are usually trained based on either a sparse prior structure over the network architecture (Richter & Wattenhofer, 2018) or based on pruning a fully-connected network to a sparse network (Yun et al., 2019; Tartaglione et al., 2018; Mocanu et al., 2018). However, it should be noted that the main objective of most sparse neural network literature has been focused on improving the memory and compute requirements while maintaining competitive accuracies compared to MLPs.

As a parallel line of research, the idea of using expressive feature combinations or groups has been suggested as a prior over the feature domain. Perhaps, the most successful and widespread use of this idea is in creating random forest models in which different trees are trained based on different feature subsets in order to deal with high-dimensional and high-variance data (Breiman, 2001). More recently, feature grouping is suggested by Aydore et al. (2019) as a statistical regularization technique to learn from datasets of large feature size and a small number of training samples. They do the forward network computation by projecting input features using samples taken from a bank of feature grouping matrices, reducing the input layer complexity and regularizing the model. In another recent study, Ke et al. (2018) used expressive feature combinations to learn from tabular datasets using a recursive encoder with a shared embedding network. They suggest a recursive architecture in which more important feature groups have a more direct impact on the final prediction.

While promising results have been reported using these methods, feature grouping has been mostly considered as a preprocessing step. For instance, Aydore et al. (2019) uses the recursive nearest agglomeration (ReNA) (Hoyos-Idrobo et al., 2018) clustering to determine feature groups prior to the analysis. Alternatively, Ke et al. (2018) defined feature groups based on a pre-trained gradient boosting decision tree (GBDT) (Friedman, 2001). Feature grouping as a preprocessing step not only increases the complexity and raises practical considerations, but also limits the optimality of the selected features in subsequent analysis. In this study, we propose an end-to-end solution to learn expressive feature groups. Moreover, we introduce a network architecture to exploit interrelations within the feature groups to reduce the network complexity and to train deeper representations.

---

[1]We plan to include a link to the source code and GitHub page related to this paper in the camera-ready version.

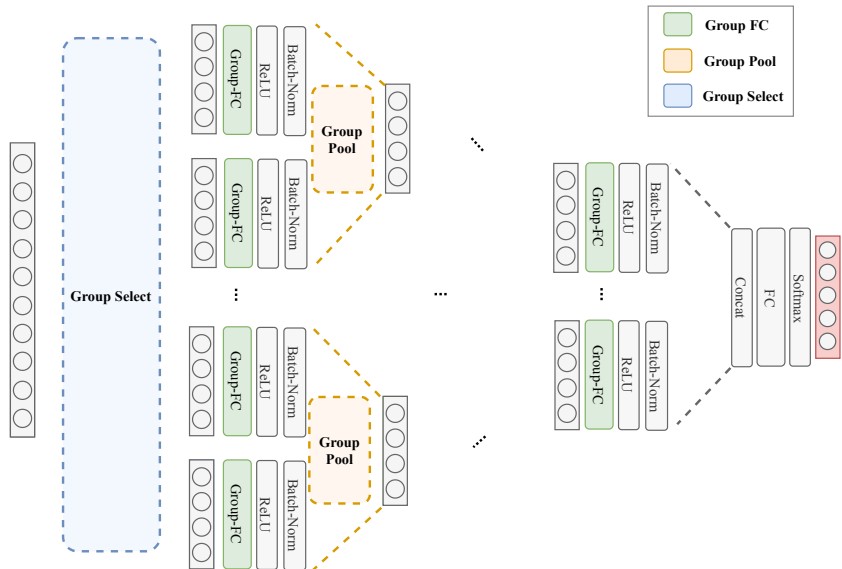

Figure 1: The GMLP network architecture.

## 3 PROPOSED METHOD

### 3.1 ARCHITECTURE OVERVIEW

In this paper, we propose GMLP which intuitively can be broken down to three stages: $(i)$ selecting expressive feature groups, $(ii)$ learning dynamics within each group individually, and $(iii)$ merging information between groups as the network grows in depth (see Figure 1). In this architecture, expressive groups are jointly selected during the training phase. Furthermore, GMLP is leveraging feature groups and using local group-wise weight layers to significantly reduce the number of parameters. While the suggested idea can be materialized as different architectures, in the current study, we suggest organization of the network as architectures resembling a binary tree spanning from leaves (i.e., features) to a certain abstraction depth closer to the root[2]. As the network grows deeper, after each local group-wise weight layer, half of the groups are merged using pooling operations, effectively reducing the width of the network while increasing the receptive field. At the last layer, all features within all groups are concatenated into a dense feature vector fed to the output layer.

### 3.2 NOTATION

We consider the generic problem of supervised classification based on a dataset of feature and target pairs, $\mathcal{D}$: $(\boldsymbol{x}_{1:N}, y_{1:N})$, where $\boldsymbol{x}_i \in \Re^d$, $y_i \in \{1 \dots C\}$, and $N$ is the number of dataset samples. Furthermore, we define group size, $m$, as the number of neurons or elements within each group, and group count, $k$, as the number of selected groups which are essentially subsets of input features. Also, $L$ is used to refer to the total depth of a network. We use $\boldsymbol{z}_i^l \in \Re^m$ to refer to activation values of group $i$ in layer $l$. In this paper, we define all vectors as column vectors.

### 3.3 NETWORK LAYERS

In this section, we present the formal definition of different GMLP network layers. The very first layer of the network, `Group-Select`, is responsible for organizing features into $k$ groups of size $m$ each. A routing matrix, $\Psi$, is used for connecting each neuron within each group to exactly one feature in the feature set:

$$\boldsymbol{z}_{1:k}^0 = \Psi \boldsymbol{x}, \tag{1}$$

---

[2]Please note that, in this paper, tree structures are considered to grow from leaves to the root. In other words, in this context, limiting the depth is synonymous with considering the tree portion spanning from a certain depth to leave nodes.

where $\Psi \in \{0, 1\}^{km \times d}$ is a sparse matrix determining features that are present in each group. As we are interested in jointly learning $\Psi$ during the training phase, we use the following continuous relaxation:

$$\Psi_{i,j} \approx \frac{\exp(\psi_{i,j}/\tau)}{\sum_{j'=1}^{j'=d} \exp(\psi_{i,j'}/\tau)}. \tag{2}$$

In this equation, $\psi$ is a real-valued matrix reparameterizing the routing matrix through a softmax operation with temperature, $\tau$. The lower the temperature, the more (2) converges to the desired discrete and sparse binary routing matrix. Note that, in the continuous relaxation, the matrix $\psi$ can be optimized via the backpropagation of gradients from classification loss terms. In the next section, we provide further detail on temperature annealing schedules as well as other techniques to enhance the $\Psi$ approximation.

Based on selected groups, we suggest local fully-connected weight layers for each group: `Group-FC`. The goal of `Group-FC` is to extract higher-level representations using the selected expressive feature subsets. This operation is usually followed by non-linearity functions (e.g., ReLU), normalization operations (e.g, Batch Norm), and dropout. Formally, `Group-FC` can be defined as:

$$\boldsymbol{z}_i^{l+1} = f(W_i^l \boldsymbol{z}_i^l + \boldsymbol{b}_i^l), \tag{3}$$

where $W_i^l \in \Re^{m \times m}$ and $\boldsymbol{b}_i^l \in \Re^m$ are the weight matrix and bias vector, applied on group $i$ at layer $l$. Here, $f$ represents other subsequent operations such as non-linearity, normalization, and dropout.

Lastly, `Group-Pool` is defined as an operation which merges representations of two groups into a single group, reducing network width by half while increasing the effective receptive field:

$$\boldsymbol{z}_i^{l+1} = pool(\boldsymbol{z}_i^l, \boldsymbol{z}_{i+k/2^{l+1}}^l), \tag{4}$$

where $\boldsymbol{z}_i^l$ and $\boldsymbol{z}_{i+k/2}^l$ are the $i$th group from the first and second halves, respectively; and $pool$ is a pooling function from $\Re^{2m}$ to $\Re^m$. In this study, we explore different variants of pooling functions such as max pooling, average pooling, or using linear weight layers as transformations from $\Re^{2m}$ to $\Re^m$. Please note that while we use a similar terminology as pooling in convolutional networks, the pooling operation explained here is not applied location-wise, but instead it is applied feature-wise, between different groups pairs.

The values of $m$ and $k$ are closely related to the number and order of feature interactions for a certain task. Using proper $m$ and $k$ values enables us to reduce the parameter space while maintaining the model complexity required to solve the task. However, finding the ideal $m$ and $k$ directly from a given dataset is a very challenging problem. In this work, we treat $m$ and $k$ as hyperparameters to be found by a hyperparameter search.

## 3.4 TRAINING

We define the objective function to be used for end-to-end training of weights as well as the routing matrix as:

$$L = -\frac{1}{N} \sum_i \sum_c y_{i,c} \log(F_\theta(\boldsymbol{x}_i)) + \lambda H(\psi) + \alpha \sum_{\omega \in \theta} ||\omega||_2^2. \tag{5}$$

In this objective function, the first term is the standard cross-entropy classification loss where $F_\theta$ denotes the GMLP network as a function with parameters $\theta$, and $N$ is the number of training samples used. The second term is an entropy loss over the distribution of the routing matrix that is weighted by the hyperparameter $\lambda$:

$$H(\psi) = -\frac{1}{d} \sum_{j=1}^{j=d} \sum_{i=1}^{i=km} \frac{\exp(\psi_{i,j})}{\sum_{j'=1}^{j'=d} \exp(\psi_{i,j'})} \log\left(\frac{\exp(\psi_{i,j})}{\sum_{j'=1}^{j'=d} \exp(\psi_{i,j'})}\right). \tag{6}$$

$H(\psi)$ is minimizing the entropy corresponding to the distribution of $\psi$ regardless of the temperature used for $\Psi$ approximation. Accordingly, $\lambda$ can be viewed as a hyperparameter and as an additional method for encouraging sparse $\Psi$ matrices. The last term in (5) is an L-2 regularization term with the hyperparameter $\alpha$ to control the magnitude of parameters in layer weights and in $\psi$. Note that

without the L-2 regularization term, $\psi$ elements may keep increasing during the optimization loop, since $\psi$ only appears in normalized form in the objective function of (5).

We use Adam (Kingma & Ba, 2014) optimization algorithm starting from the default 0.001 learning rate and reducing the learning rate by a factor of 5 as the validation accuracy stops improving. Regarding the temperature annealing, during the training, the temperature is exponentially decayed from 1.0 to 0.01. In order to initialize the `Group-FC` weights, we used Xavier initialization (Glorot & Bengio, 2010) with m for both fan-in and fan-out values. Similarly, the $\psi$ matrix is initialized by setting the fan-in equal to $d$ and fan-out to $km$.

Further detail on architectures and hyperparameters used for each specific experiment as well as details on the software implementation are provided as appendices to this paper.

### 3.5 ANALYSIS

The computational complexity of GMLP at the prediction time can be written as (for simplicity, ignoring bias and pooling terms):

$$km + km^2 + \frac{km^2}{2} + \frac{km^2}{4} + ... + \frac{km^2}{2^{L-1}} + C\frac{km}{2^{L-1}}. \tag{7}$$

In this series, the first term, $km$, is the work required to organize features to groups. The subsequent terms, except the last term, are representing the computational cost of local fully-connected operations at each layer. The last term is the complexity of the output layer transformation from the concatenated features to the number of classes. Therefore, the computational complexity of GMLP at the prediction time can be written as $\mathcal{O}(km^2 + \frac{Ckm}{2^{L-1}})$. In comparison, the computational complexity of an MLP with a similar network width would be:

$$kmd + k^2m^2 + \frac{k^2m^2}{2} + \frac{k^2m^2}{4} + ... + \frac{k^2m^2}{2^{L-1}} + C\frac{km}{2^{L-1}}, \tag{8}$$

where the first term is the work required for the first network layer from $d$ to $km$ neurons, the second term is corresponding to a hidden layer of size $km$, and so forth. The last term is the complexity of the output layer similar to the case of GMLP. The overall work required from this equation is of $\mathcal{O}(kmd + k^2m^2 + \frac{Ckm}{2^{L-1}})$ complexity. This is substantially higher than GMLP, for typical $k$, $d$, and $C$ values.

Additionally, the density of the `Group-FC` layer connections can be calculated as: $\frac{km^2}{k^2m^2} = \frac{1}{k}$, which is very small for reasonably large number of $k$ values used in our experiments. Also, assuming pooling operations in every other layer, the receptive field size or the maximum number of features impacting a neuron at layer $l$ can be written as $2^{l-1}m$. For instance, a neuron in the first layer of the network is only connected to $m$ features, and a neuron in the second layer is connected to two groups or $2m$ features and so forth.

## 4 EXPERIMENTS

### 4.1 EXPERIMENTAL SETUP

The proposed method is evaluated on five different real-world datasets, covering various domains and applications: permutation invariant CIFAR-10 (Krizhevsky et al., 2009), human activity recognition (HAPT) (Anguita et al., 2013), diabetes classification (Kachuee et al., 2019), UCI Landsat (Dua & Graff, 2017), and MIT-BIH arrhythmia classification (Moody & Mark, 2001) datasets. Additionally, we use MNIST (LeCun et al., 2010) and a synthesized dataset to provide further insight into the operation of GMLP (see Section 4.4). Table 1 presents a summary of datasets used in this study. Regarding the CIFAR-10 dataset, we permute the image pixels to discard pixel coordinates in our experiments. Note that the permutation is not changing across samples, it is merely a fixed random ordering used to remove pixel coordinates for each experiment. For all datasets, basic statistical normalization with $\mu = 0$ and $\sigma = 1$ is used to normalize features as a preprocessing step. The only exception is CIFAR-10 for which we used the standard channel-wise normalization and standard data augmentation (i.e., random crops and random horizontal flips). The standard test and train data splits were used as dictated by dataset publishers. In cases that the separated sets are not provided,

Table 1: Summary of datasets used in our experiments.

| Dataset | # Train Samples | # Test Samples | # Features | # Classes | Domain |
|---|---|---|---|---|---|
| **CIFAR-10**[a] (Krizhevsky et al., 2009) | 50,000 | 10,000 | 3072 | 10 | Image Classification |
| **HAPT** (Anguita et al., 2013) | 6,002 | 2,451 | 561 | 5 | Activity Recognition |
| **Diabetes** (Kachuee et al., 2019) | 73,649 | 18,413 | 45 | 3 | Disease Diagnosis |
| **Landsat** (Dua & Graff, 2017) | 4,435 | 2,000 | 36 | 6 | Satellite Imaging |
| **MIT-BIH** (Moody & Mark, 2001) | 87,554 | 21,892 | 187 | 5 | Time Series |
| **MNIST** (LeCun et al., 2010) | 60,000 | 10,000 | 784 | 10 | Digit Classification |
| **Synthesized** (Section 4.4) | 5,120 | 1,280 | 6 | 2 | See Figure 8 |

[a]Permuted version, i.e. pixel coordinates are ignored.

test and train subsets are created by randomly splitting samples to $20\%$ for test and the rest for training/validation.

We compare the performance of the proposed method with recent related work including Self-Normalizing Neural Networks (SNN) (Klambauer et al., 2017), Sparse Evolutionary Training (SET) (Mocanu et al., 2018)[3], Feature Grouping as a Stochastic Regularizer (in this paper, denoted as FGR) (Aydore et al., 2019)[4] as well as the basic dropout regularized and batch normalized MLPs. In order to ensure a fair comparison, we adapted source codes provided by other work to be compatible with our data loader and preprocessing modules.

Furthermore, for each method, we conducted an extensive hyperparameter search using Microsoft Neural Network Intelligence (NNI) toolkit[5] and the Tree-structured Parzen Estimator (TPE) tuner (Bergstra et al., 2011) covering different architectural and learning hyperparameters for each case. More detail on hyperparameter search spaces and specific architectures used in this paper is provided in Appendix A and Appendix B. We run each case using the best hyperparameter configuration eight times and report mean and standard deviation values.

## 4.2 RESULTS

Table 2 presents a comparison between the proposed method (GMLP) and 4 other baselines: MLP, SNN (Klambauer et al., 2017), SET (Mocanu et al., 2018), and FGR (Aydore et al., 2019). As it can be seen from this comparison, GMLP outperforms other work, achieving state-of-the-art classification accuracies. Concerning the CIFAR-10 results, to the best of our knowledge, GMLP achieves a new state-of-the-art performance on permutation invariant CIFAR-10 augmented using the standard data augmentation. We believe that leveraging expressive feature groups enables GMLP to consistently perform better across different datasets.

To compare model complexity and performance we conduct an experiment by changing the number of model parameters and reporting the resulting test accuracies. Here, we reduce the number of parameters by reducing the width of each network; i.e. reducing the number of groups and hidden neurons for GMLP and MLP, respectively. Figure 2 shows accuracy versus the number of parameters for the GMLP and MLP baseline on CIFAR-10 and MIT-BIH datasets. Based on this figure, GMLP is able to achieve higher accuracies using significantly less number of parameters. It is consistent with the complexity analysis provided in Section 3.5. Note that in this comparison, we consider the number of parameters involved at the prediction time.

## 4.3 ABLATION STUDY

Figure 3 presents an ablation study comparing the performance of GMLP on CIFAR-10 dataset for networks trained: $(i)$ using both the temperature annealing and the entropy loss objective, $(ii)$ using only temperature annealing without the entropy loss objective, $(iii)$ using no temperature annealing but using the entropy loss objective, $(iv)$ not using any of the temperature annealing or the entropy loss

---

[3]https://github.com/dcmocanu/sparse-evolutionary-artificial-neural-networks
[4]https://github.com/sergulaydore/Feature-Grouping-Regularizer
[5]https://github.com/microsoft/nni

Table 2: Comparison of top-1 test accuracies for GMLP and other work.

| Dataset | Accuracy (%) | | | | |
|---|---|---|---|---|---|
| | **GMLP** | **MLP** | **SNN**[a] | **SET**[b] | **FGR**[c] |
| **CIFAR-10**[d] (Krizhevsky et al., 2009) | **73.76** (±0.14) | 68.15 (±0.56) | 66.88 (±0.30) | 72.71 (±0.29) | 45.90 (±0.40) |
| **HAPT** (Anguita et al., 2013) | **96.34** (±0.19) | 95.73 (±0.38) | 95.47 (±0.09) | 71.35 (±0.74) | 91.57 (±0.31) |
| **Diabetes** (Kachuee et al., 2019) | **90.92** (±0.07) | 90.68 (±0.06) | 90.31 (±0.10) | 85.07 (±0.61) | 89.24 (±0.08) |
| **Landsat** (Dua & Graff, 2017) | **91.54** (±0.16) | 91.21 (±0.44) | 91.37 (±0.21) | 91.03 (±0.56) | 90.70 (±0.17) |
| **MIT-BIH** (Moody & Mark, 2001) | **98.74** (±0.04) | 98.65 (±0.01) | 98.56 (±0.02) | 98.10 (±0.01) | 98.13 (±0.03) |

[a]Self-Normalizing Neural Networks (Klambauer et al., 2017)
[b]Sparse Evolutionary Training (Mocanu et al., 2018)
[c]Feature Grouping as a Stochastic Regularizer (Aydore et al., 2019)
[d]Permuted version, i.e. pixel coordinates are ignored.

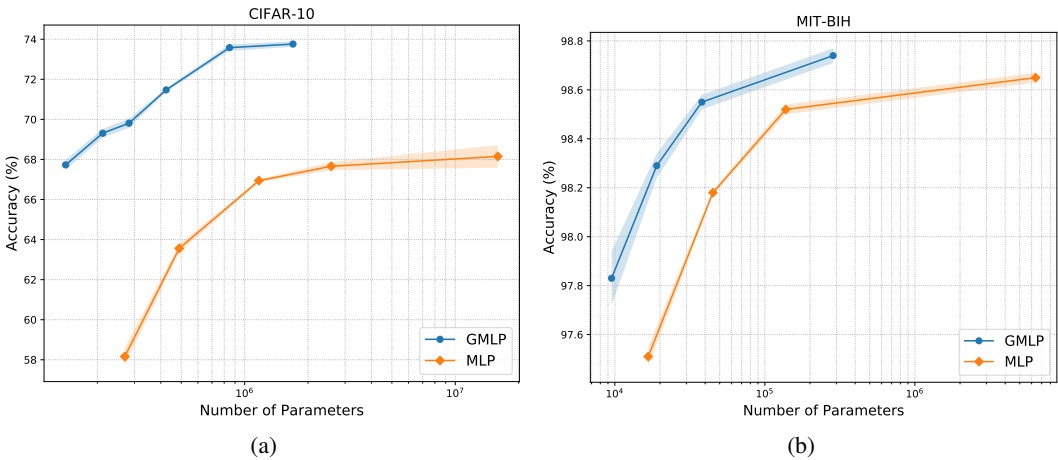

Figure 2: Accuracy versus number of parameters for this work (GMLP) and the MLP baseline: (a) CIFAR-10 dataset, (b) MIT-BIH dataset. The x-axis is in a logarithmic scale

objective. From this figure, it can be seen that excluding both techniques leads to a significantly lower performance. However, using any of the two techniques leads to relatively similar high accuracies. It is consistent with the intuition that the functionality of these techniques is to encourage learning sparse routing matrices, either using softmax temperatures or entropy regularization to achieve this. In this paper, in order to ensure sparse and low complexity routing matrices, we use both techniques simultaneously as in case $(i)$.

Figure 4 shows a comparison between GMLP models trained on CIFAR-10 using different pooling types: $(i)$ linear transformation, $(ii)$ max pooling, and $(iii)$ average pooling. As it can be seen from this comparison, while there are slight differences in the convergence speed of using different pooling types, all of them achieve relatively similar accuracies. In our experiments, we decided to use max pooling and average pooling as they provide reasonable results without the need to introduce additional parameters required for the linear pooling method.

Figure 5 shows learning curves for training CIFAR-10 GMLP models using different group sizes. As it can be seen from this figure, using very small group sizes would cause a reduction in the final accuracy. At the other extreme, the improvement achieved using larger values is negligible for $m$ values more than 16. Finally, Figure 6 shows a comparison between learning curves for using a different number of groups. Using very small $k$ values result in a significant reduction in performance. However, the rate of performance gains for using more groups is very small for $k$ of more than 1536. Note that the number of model parameters and compute scales linearly with $k$ and quadratically with $m$ (see Section 3.5).

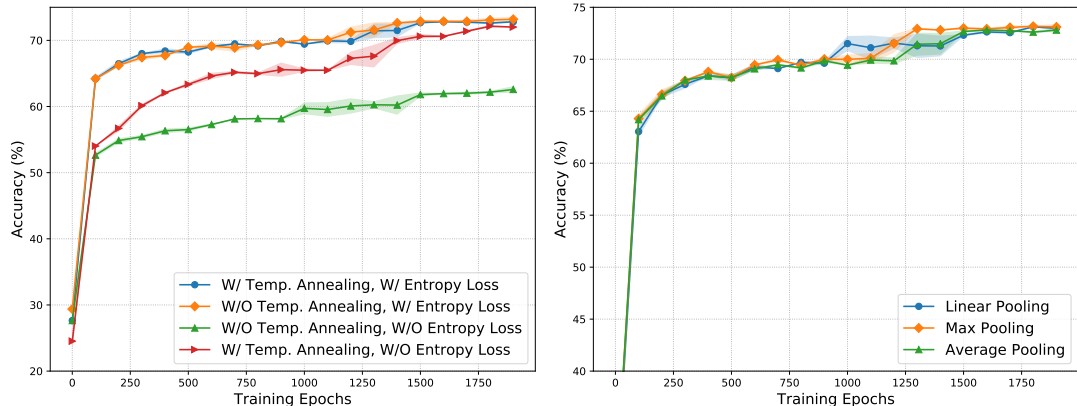

Figure 3: Ablation study demonstrating the impact of temperature annealing and entropy loss terms.

Figure 4: Ablation study demonstrating the impact of different pooling functions.

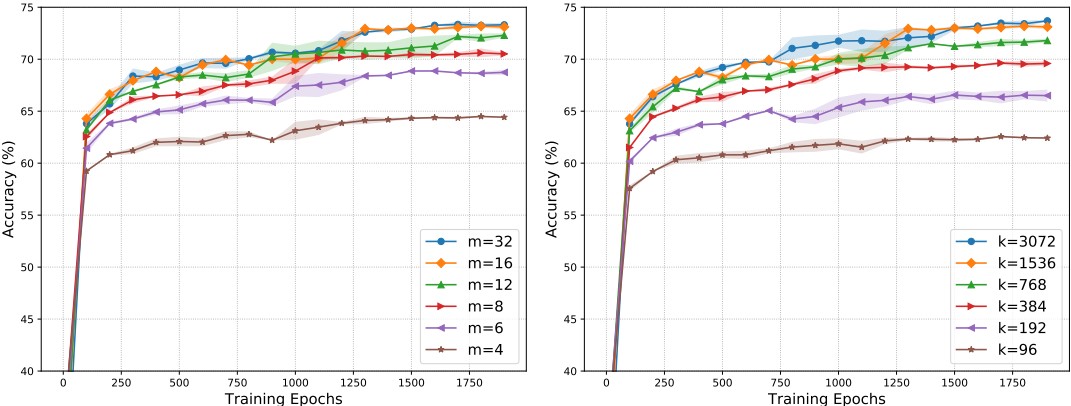

Figure 5: Ablation study on the impact of using different group sizes ($m$). For this experiment, we used $k$=1536.

Figure 6: Ablation study on the impact of using different number of groups ($k$). For this experiment, we used $m$=16.

## 4.4 EXPERIMENTS ON MNIST AND SYNTHESIZED DATA

MNIST dataset (LeCun et al., 2010) is used to visually inspect the performance of the `Group-Select` layer. Figure 7 shows a heat-map of how frequently each pixel is selected across all feature groups for: ($a$) original MNIST samples, ($b$) MNIST samples where the lower-half is replaced by Gaussian noise. From Figure 7a, it can be seen that most groups are selecting pixels within the center of the frame, effectively discarding margin pixels. This is consistent with other work which show the importance of different locations for MNIST images (Kachuee et al., 2018). Apart from this, in Figure 7b, a version of the MNIST dataset is used in which half of the frame does not provide any useful information for the downstream classification task. From this figure, GMLP is not selecting any features to be used from the lower region.

In order to show the effectiveness of GMLP, we synthesized a dataset which has intrinsic and known expressive feature groups. Specifically, we used a simple Bayesian network as depicted in Figure 8. This network consists of six binary features, A to F, interacting with each other as specified by the graph edges, which determine the distribution of the target node, J. The graph and conditionals are designed such that each of the nodes in the second level take the XOR value of their parents with a 99% probability. The target node, J, is essentially one with a high probability if at least two of the second level nodes are one. We synthesized dataset by sampling 6,400 samples from the network (1,280 samples for test and the rest of training/evaluation). On this dataset, we trained a very simple GMLP consisting of four groups of size two, one group-wise fully-connected layer, and an output layer. Figure 9 shows the features selected for each group after the training phase (i.e., the $\Psi$ matrix).

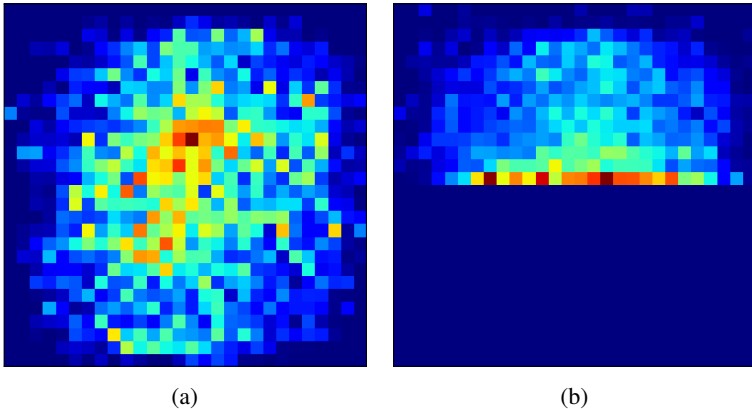

(a)                                                   (b)

Figure 7: MNIST visualization of pixels selected by the `Group-Select` layer: (a) using complete images as input, (b) using images that the lower half is replaced by Gaussian noise. In this figure, warmer colors represent pixels being being present in more groups.

From this figure, the `Group-Select` layer successfully learns to detect the feature pairs that are interacting, enabling the `Group-FC` layers to decode the non-linear XOR relations.

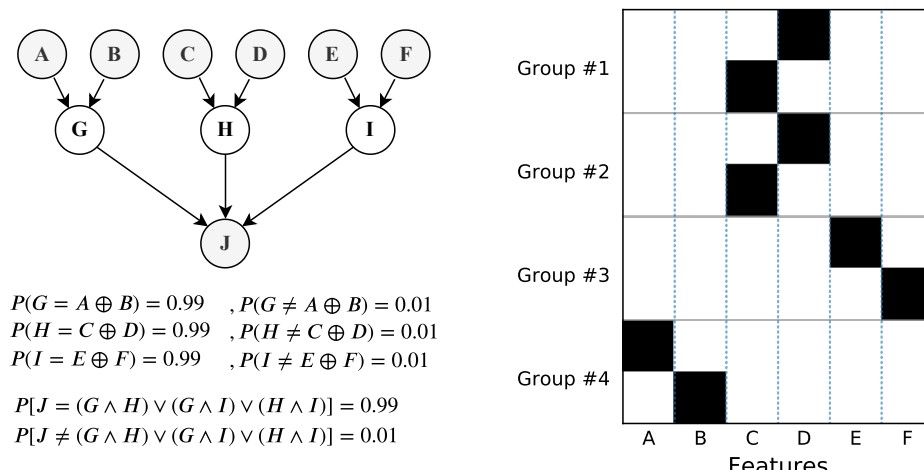

$P(G = A \oplus B) = 0.99 \quad , P(G \neq A \oplus B) = 0.01$
$P(H = C \oplus D) = 0.99 \quad , P(H \neq C \oplus D) = 0.01$
$P(I = E \oplus F) = 0.99 \quad , P(I \neq E \oplus F) = 0.01$

$P[J = (G \land H) \lor (G \land I) \lor (H \land I)] = 0.99$
$P[J \neq (G \land H) \lor (G \land I) \lor (H \land I)] = 0.01$

Figure 8: The Bayesian network and conditionals used to generate the synthesized dataset of binary features A-F and target J.

Figure 9: Visualization of the selected features within each group. Every two consecutive rows show features selected for a certain group.

## 5 DISCUSSION

Intuitively, training a GMLP model with certain groups can be viewed as a prior assumption over the number and order of interactions between the features. It is a reasonable prior assumption as in many natural datasets, a conceptual hierarchy exists where only a limited number of features interact with each other. Furthermore, it is consistent with the discoveries made in understanding the human decision-making process; finding that we are only able to consider at most nine factors at the same time during a decision making process (Cowan, 2001; Baddeley, 1994).

Additionally, GMLP can be considered as a more general neural counterpart of random forests. Both models use different subsets of features (i.e., groups) and learn interactions within each group. One major difference between the two methods is the fact that GMLP combines information between different groups using pooling operations, while random forest uses the selected features to train an

ensemble of independent trees on each group. From another perspective, the idea of studying feature groups is closely related to causal models such as Bayesian networks and factor graphs (Darwiche, 2009; Neapolitan et al., 2004; Clifford, 1990). These methods are often impractical for large-scale problems, because without a prior over the causal graph, they require an architecture search of the NP-complete complexity or more.

## 6 CONCLUSION

In this paper, we proposed GMLP as a solution for deep learning in domains where the feature interactions are not known as prior and do not admit the use of convolutional or other techniques leveraging domain priors. GMLP jointly learns expressive feature combinations and employs group-wise operations to reduce the network complexity. We conducted extensive experiments demonstrating the effectiveness of the proposed idea and compared the achieved performances with state-of-the-art methods in the literature.

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

## A  HYPERPARAMETER SEARCH SPACE

Tables 3-5 present the hyperparameter search space considered for experiments on GMLP, MLP, SNN, and FGR, respectively. For the GMLP search space, the number of groups is adjusted based on the number of features and samples in each specific task. Also, the number of layers is adjusted to be compatible with the number of groups being used. Regarding the FGR experiments, due to scalability issues of the published source provided by the original authors, we were only able to train networks with at most two hidden layers. For SET, as their architecture is evolutionary i.e., prunes certain weights and adds new ones, we only explored using a different number of hidden neurons in the range of 500 to 4000.

Regarding the number of epochs, we used 2000 epochs for CIFAR-10, 1000 epochs for HAPT, 100 epochs for Diabetes, 300 epochs for Landsat, and 300 epochs for MIT-BIH experiments. The only exception is the SNN experiments where we had to reduce the learning rate to increase the stability of the training resulting in more epochs required to converge.

Table 3: Hyperparameter search space used for GMLP experiments.

| Hyperparameter | Considered Values |
|---|---|
| Number of hidden layers | $\{1, 2, ..., 8\}^a$ |
| Number of groups | $16 - 4096^b$ |
| Size of groups | $4 - 32$ |
| Lambda | $[10^{-2}, 10^4]$ |
| Alpha | $[10^{-12}, 10^{-1}]$ |
| Dropout rate | $[0, 1]$ |

[a]The range is adjusted based on the number of groups.
[b]The range is adjusted based on the number of features and sample size.

Table 4: Hyperparameter search space used for MLP and SNN experiments.

| Hyperparameter | Considered Values |
|---|---|
| Number of hidden layers | $\{1, 2, ..., 6\}$ |
| Size of hidden layers | $[0.05 \times n_{features}] - [20 \times n_{features}]$ |
| Alpha | $[10^{-12}, 10^{-1}]$ |
| Dropout rate | $[0, 1]$ |

Table 5: Hyperparameter search space used for FGR experiments.

| Hyperparameter | Considered Values |
|---|---|
| Number of hidden layers | $\{1, 2\}$ |
| Size of hidden layers | $n_{features} - [50 \times n_{features}]$ |
| Number of groups | $2 - n_{features}$ |
| Alpha | $[10^{-12}, 10^{-1}]$ |

# B ARCHITECTURES

Table 6,7,8,9,10 show the selected architectures for GMLP, MLP, SNN, SET, and FGR, respectively. We used the following notation to indicate different layer types and parameters: `GSel-k-m` represents a `Group-Select` layer selecting `k` groups of `m` features each. `GFC` indicates `Group-FC` layers, and `FC-x` represents fully-connected layer with `x` hidden neurons. `GPool-x` is a `Group-Pool` layer of type `x` (max, mean, linear, etc.). `Concat` is concatenation of groups used prior to the output layer in GMLP architectures. `SC-x` refers to SET sparse evolutionary layer of size `x`.

Table 6: GMLP architectures used in our experiments.

| Dataset | Architecture |
|---|---|
| **CIFAR-10** | `GSel-1536-16, GFC, ReLU, BNorm, GPool-max, GFC, ReLU, BNorm, GPool-max, GFC, ReLU, BNorm, GPool-max, GFC, ReLU, BNorm, Concat, FC-10, Softmax` |
| **HAPT** | `GSel-288-12, GFC, ReLU, BNorm, GPool-mean, GFC, ReLU, BNorm, Concat, FC-5, Softmax` |
| **Diabetes** | `GSel-16-16, GFC, ReLU, BNorm, GPool-mean, GFC, ReLU, BNorm, GPool-mean, GFC, ReLU, BNorm, GPool-mean, GFC, ReLU, BNorm, GPool-mean, Concat, FC-3, Softmax` |
| **Landsat** | `GSel-88-16, GFC, ReLU, BNorm, GPool-mean, GFC, ReLU, BNorm, GPool-mean, GFC, ReLU, BNorm, GPool-mean, Concat, FC-6, Softmax` |
| **MIT-BIH** | `GSel-240-24, GFC, ReLU, BNorm, GPool-mean, GFC, ReLU, BNorm, GPool-mean, GFC, ReLU, BNorm, GPool-mean, Concat, FC-5, Softmax` |
| **MNIST** | `GSel-64-16, GFC, ReLU, BNorm, GPool-mean, GFC, ReLU, BNorm, GPool-mean, GFC, ReLU, BNorm, GPool-mean, GFC, ReLU, BNorm, Concat, FC-10, Softmax` |
| **Synthesized** | `GSel-4-2, GFC, ReLU, BNorm, Concat, FC-2, Softmax` |

Table 7: MLP architectures used in our experiments.

| Dataset | Architecture |
|---|---|
| **CIFAR-10** | `FC-3072, ReLU, BNorm, FC-2764, ReLU, BNorm, FC-2488, ReLU, BNorm, FC-10, Softmax` |
| **HAPT** | `FC-106, ReLU, BNorm, FC-21, ReLU, BNorm, FC-5, Softmax` |
| **Diabetes** | `FC-45, ReLU, BNorm, FC-45, ReLU, BNorm, FC-45, ReLU, BNorm, FC-45, ReLU, BNorm, FC-3, Softmax` |
| **Landsat** | `FC-68, ReLU, BNorm, FC-68, ReLU, BNorm, FC-68, ReLU, BNorm, FC-68, ReLU, BNorm, FC-6, Softmax` |
| **MIT-BIH** | `FC-1737, ReLU, BNorm, FC-1737, ReLU, BNorm, FC-1737, ReLU, BNorm, FC-5, Softmax` |

Table 8: SNN architectures used in our experiments.

| Dataset | Architecture |
|---------|-------------|
| CIFAR-10 | `FC-3901, SeLU, BNorm, FC-3901, SeLU, BNorm,`
`FC-3901, SeLU, BNorm, FC-10, Softmax` |
| HAPT | `FC-510, ReLU, BNorm, FC-510, SeLU, BNorm,`
`FC-510, SeLU, FC-5, Softmax` |
| Diabetes | `FC-105, SeLU, BNorm, FC-105, SeLU, BNorm,`
`FC-105, SeLU, BNorm, FC-3, Softmax` |
| Landsat | `FC-816, SeLU, BNorm, FC-816, SeLU, BNorm, FC-6, Softmax` |
| MIT-BIH | `FC-1140, SeLU, BNorm, FC-1140, SeLU, BNorm,`
`FC-1140, SeLU, BNorm, FC-5, Softmax` |

Table 9: SET architectures used in our experiments.

| Dataset | Architecture |
|---------|-------------|
| CIFAR-10 | `SC-4000, SReLU, SC-1000, SReLU, SC-4000, SReLU, FC-10, Softmax` |
| HAPT | `SC-500, SReLU, SC-500, SReLU, SC-500, SReLU, FC-5, Softmax` |
| Diabetes | `SC-1000, SReLU, SC-1000, SReLU, SC-1000, SReLU, FC-3, Softmax` |
| Landsat | `SC-1000, SReLU, SC-1000, SReLU, SC-1000, SReLU, FC-6, Softmax` |
| MIT-BIH | `SC-1000, SReLU, SC-1000, SReLU, SC-1000, SReLU, FC-5, Softmax` |

Table 10: FGR architectures used in our experiments.

| Dataset | Architecture |
|---------|-------------|
| CIFAR-10 | `Group-256, FC-3072, ReLU, FC-10, Softmax` |
| HAPT | `Group-104, FC-12173, ReLU, FC-5, Softmax` |
| Diabetes | `Group-44, FC-337, ReLU, FC-3, Softmax` |
| Landsat | `Group-32, FC-1577, ReLU, FC-6, Softmax` |
| MIT-BIH | `Group-160, FC-3444, ReLU, FC-5, Softmax` |

## C    SOFTWARE IMPLEMENTATION

Table 11 presents the list of software dependencies and versions used in our implementation. To produce results related to this paper, we used a workstation with 4 NVIDIA GeForce RTX-2080Ti GPUs, a 12 core Intel Core i9-7920X processor, and 128 GB memory. Each experiment took between about 30 minutes to 72 hours, based on the task and method being tested.

Table 11: Software dependencies.

| Dependency | Version |
|---|---|
| python | 3.7.1 |
| pytorch | 1.1.0 |
| torchvision | 0.2.1 |
| cuda100 | 1.0 |
| ipython | 6.5.0 |
| jupyter | 1.0.0 |
| numpy | 1.15.4 |
| nni | 0.9.1.1 |
| pandas | 0.23.4 |
| scikit-learn | 0.19.2 |
| scipy | 1.1.0 |
| pomegranate | 0.11.1 |
| tqdm | 4.32.1 |
| matplotlib | 3.0.1 |

# D  VISUAL ANALYSIS

In Figure 10, we present a visualization of the selected feature for 25 randomly selected groups in our final CIFAR-10 architecture. Red, green, and blue colors indicate which channel is selected for each location. Compared to visualizations that are frequently used for convolutional networks, as GMLP has the flexibility to select pixels at different locations and different color channels, it is not easy to find explicit patterns in this visualization. However, one noticeable pattern is that features selected from a certain color channel usually appear in clusters resembling irregularly shaped patches.

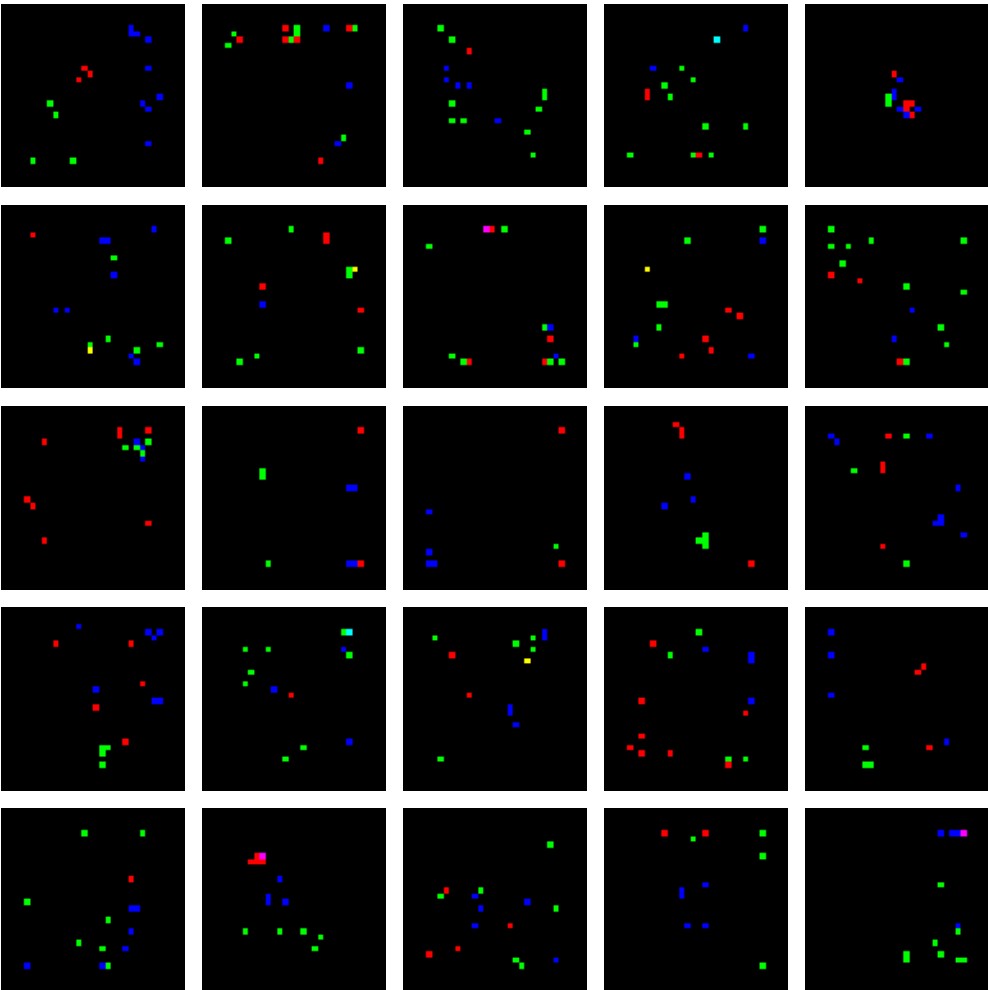

Figure 10: Visualization of pixels selected by each group for the CIFAR-10 GMLP architecture. Red, green, and blue colors indicate which channel is selected for each location. Due to space limitations, 25 random groups out of 1536 total groups visualized here.

Figure 11 shows the frequency in which each CIFAR-10 location is selected by the GMLP network. From this visualization, GMLP is mostly ignoring the border areas which can be a result of the data augmentation process used to train the network i.e., randomly cropping the center area and padding the margins.

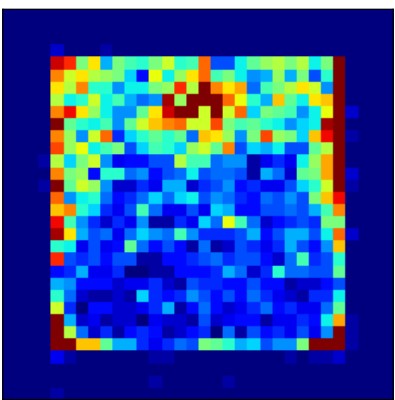

Figure 11: Visualization of pixels selected by the `group-select` layer for the CIFAR-10 GMLP model. Warmer colors represent features that are being selected more frequently.

