# OpenReview forum: "Group-Connected Multilayer Perceptron Networks"
_ICLR.cc/2020/Conference — Reject_

### Official Review · AnonReviewer1 · 2019-10-24
**Official Blind Review #1**

**Rating:** 3

**Review:**

This paper studies supervised classification problems where features are unstructured. For these problems, the authors propose a new neural network architecture that first reorganize the features into groups, then builds feed-forward networks on top each group, and finally aggregate the hidden nodes of each group to produce the final output. Empirical and ablation studies are conducted to show the performance of this approach.

My detailed comments are as follows.

1. The intuition of this approach should be better explained. In equation (1) the features are group together using a binary matrix. Then the authors suggest using a relaxed version in equation (2) involving softmax function. What is the intuition here? If the feature is very high-dimensional, it seems that the normalization factor in equation (2) might be hard to compute. Moreover, what is the relationship between $k$, $m$, and $d$?

2. It seems that the neural network architecture is a simple variation of the standard MLP, except that the bottom layer is changed to a linear layer $ z = \Psi x$, where $\Psi$ is defined using $\{ \psi_{ij}, i \in [km], j\in [d] \} $ and a softmax operation. It seems that the contribution of the network structure is rather incremental.

3. In terms of the experiments, it seems that the results are very similar to that of the MLP, although slightly better. Moreover, the datasets used seem small, with no more than 10^6 data points in all datasets. The largest dataset is the Permutation invariant CIFAR-10, which has 50000 data with 3072 features. It would be interesting to see how this method works for high-dimensional datasets where the number of features is large.



**Experience Assessment:**

I have published one or two papers in this area.

**Review Assessment: Checking Correctness Of Derivations And Theory:**

I carefully checked the derivations and theory.

**Review Assessment: Checking Correctness Of Experiments:**

I carefully checked the experiments.

**Review Assessment: Thoroughness In Paper Reading:**

I read the paper thoroughly.

---

> ### Author Response · Authors · 2019-11-14
> **To Reviewer #1 (2/2)**
>
>
> *Comment: “3. In terms of the experiments, it seems that the results are very similar to that of the MLP, although slightly better. Moreover, the datasets used seem small, with no more than 10^6 data points in all datasets. The largest dataset is the Permutation invariant CIFAR-10, which has 50000 data with 3072 features. It would be interesting to see how this method works for high-dimensional datasets where the number of features is large.”
>
> Regarding the comparison of results with MLP, on CIFAR-10 we have more than 5% improvement over MLP which we believe is a new state-of-the-art for architectures not using pixel coordinates. For the rest of the datasets which are health datasets or UCI benchmarks, even a small percentage of improvement is often very important and frequently used in literature to compare the performance of different methods. From the results presented in Table 2, GMLP is consistently providing more accurate results compared to all other work which we believe is significant given the diversity of the tasks compared here.
>
> Moreover, as demonstrated in Figure 2, the proposed architecture is much more efficient in terms of the number of parameters or model complexity at the prediction time. In other words, GMLP uses an order of magnitude less parameters and still achieves better results compared to the best MLP.
>
> We agree with the reviewer that the evaluation of the suggested method on larger datasets would be interesting. In the current version, we used 7 different datasets where 5 of which were used to compare with the related work. Our largest task, CIFAR-10, is arguably the largest dataset that other competitors (cited in the related work or in the comparison of results) used for their evaluation. Note that in the literature related to this work, UCI benchmarks, MNIST, etc. are most frequently used for the evaluation.

---

> ### Author Response · Authors · 2019-11-14
> **To Reviewer #1 (1/2)**
>
>
> Thank you for reviewing the manuscript and helpful comments. Please find a point-to-point response to your comments in the following.
>
> -------------------------------------------
> *Comment: “1. The intuition of this approach should be better explained. In equation (1) the features are group together using a binary matrix. Then the authors suggest using a relaxed version in equation (2) involving softmax function. What is the intuition here? If the feature is very high-dimensional, it seems that the normalization factor in equation (2) might be hard to compute. Moreover, what is the relationship between m, k, and d?”
>
> Equation (2) is a continuous relaxation used to make the matrix multiplication operation in Equation (1) differentiable. Here, a softmax function with temperature is used for this purpose.
>
> The use of softmax functions for the continuous relaxation of discrete parameters is frequent in the variational learning literature. Intuitively, the softmax function guarantees that each row in the $\psi$ matrix is a valid probability distribution. The temperature is annealed during the training process so that at the very end of the training process the relaxed distribution converges to a discrete distribution. Therefore, the relaxation helps us in turning the discrete objective into a continuous problem. Otherwise, the discrete optimization problem would have been very complex and impractical to be used for end-to-end learning.
>
> Based on our initial experiments, the temperature annealing does not require any extensive adjustment for different experiments and works effectively for high-dimensional problems as long as a reasonable annealing function is used from high temperatures (e.g., 1) to low temperatures (e.g., 0.001).
>
>
> Regarding the relationship between m, k, d:
> m is determining the size of each group for a GMLP architecture
> k is determining the number of groups a GMLP architecture
> d is the number of features for a certain task
>
> Ideally, the values of m and k are related to the number and order of feature interactions for a certain task. Using proper m and k values enables us to reduce the parameter space while maintaining the model complexity required to solve the task. However, finding the ideal m and k directly from a given dataset is a very challenging problem. In this work, we treat m and k as hyperparameters to be found by a hyperparameter search on a validation set, similar to what we have for other network/training hyperparameters. See Appendix A for hyperparameter search spaces and Appendix B for selected architectures for each task.
>
> Based on our experiments, usually m<<d, and m*k has relatively the same order (in some cases, slightly more) as d. For instance, for the CIFAR-10 architecture: m=16, k=1536, and d=3072.
>
> To address your comment and to clarify for our readers, we added the discussion above to the last paragraph of Section 3.3 in the revised version.
>
> -------------------------------------------
> *Comment: “2. It seems that the neural network architecture is a simple variation of the standard MLP, except that the bottom layer is changed to a linear layer $z=\Psi x$, where $\Psi$ is defined using ${\Psi_{i.j|i \in [km], j \in [d]}}$ and a softmax operation. It seems that the contribution of the network structure is rather incremental.”
>
> The grouping idea at the first layer of the network is one of the main contributions of this paper; however, we believe that the binary-tree like architecture is an equally important contribution. The suggested architecture learns how to utilize the selected groups to control the network complexity. Indeed, we believe that the combination of the first and the second contributions mentioned above is making this paper outstanding compared to the current literature in sparse network architecture and network architecture using expressive feature groups.

---

### Official Review · AnonReviewer3 · 2019-11-04
**Official Blind Review #3**

**Rating:** 3

**Review:**

This paper addresses the problem of learning expressive feature combinations in order to improve learning for domains where there is no known structure between features. These settings would normally lead to the use of fully-connected MLP networks, which unfortunately have problems with efficient training and generalization after a few layers of depth. The main idea is to use grouping at first, in combination with smaller fully-connected layers for each group, as well as pooling pairs of groups together as the layers go on. Results are shown as comparisons on 5 real-world datasets, and intuitive visualizations on two other datasets. Related work covered MLPs, regularization techniques, sparse networks, random forest models, and other feature grouping. The paper is well written and easy to read. This work did a good job with giving implementation details as well as performing hyperparameter searches and giving the baselines a good effort.

My current decision is a weak reject, for a well-written paper, but some concerns as follows:
-The results do not show much improvement (i.e., < 0.3% improvement for 3 of the datasets, and < 1% for another one), aside from CIFAR-10. Considering that the premise of the paper is that MLP’s are not good enough when dealing with data in which the relationships between features are unknown, it seems like these are definitely not good datasets on which to demonstrate this notion of “there has been little progress in deep reinforcement learning for domains without a known structure between features.”
-The MNIST visualization of group-select felt informative, but the XOR example for grouping visualizations seemed too easy. It would’ve been good to see visualizations or intuitions regarding grouping for harder datasets, in order to be convinced of the need for more expressive feature representations than standard MLP’s.
-I’m not an expert on causality, but it seems like citations from that area are required for this problem statement of dealing with features where the connections between them are unknown but potentially very important.

Less major:
-It would have been nice to include related work on other ways to encourage inter-feature interactions, such as perhaps taking the outer product of the input with itself.
-It seems like different sizes per group would be a more realistic expectation, and that perhaps this should be worked into the algorithm. Similarly, pooling only 2 groups together (from pre-specified positions) seems like it would be limiting as well. It also seems like the algorithm should account for being able to use a high-level feature from one layer as part of multiple groups in the future (i.e. reuse). Even if any of these options don’t make a difference, it would be good to check/evaluate.

Minor:
-Equation 8 did not fully make sense to me.
-Why were “random horizontal flips” used as preprocessing for the permutation-invariant CIFAR-10 dataset? This shouldn’t make a difference at all if the pixels become randomly shuffled anyway.

**Experience Assessment:**

I have read many papers in this area.

**Review Assessment: Checking Correctness Of Derivations And Theory:**

I assessed the sensibility of the derivations and theory.

**Review Assessment: Checking Correctness Of Experiments:**

I carefully checked the experiments.

**Review Assessment: Thoroughness In Paper Reading:**

I read the paper thoroughly.

---

> ### Author Response · Authors · 2019-11-14
> **To Reviewer #3 (2/2)**
>
>
> **Less major:
> -------------------------------------------
> *Comment: “It would have been nice to include related work on other ways to encourage inter-feature interactions, such as perhaps taking the outer product of the input with itself.”
>
> In Section 2 (related work), the last two paragraphs are dedicated to the discussion of other work using the feature grouping idea.
> For instance, we discuss Aydore et al. (2019) which uses feature groups to regularize the network training, or Ke et al. (2018) which exploits expressive feature combinations extracted from a GBDT to learn from tabular data. Please let us know if you have any specific paper in mind and we would be happy to discuss that one as well.
>
> -------------------------------------------
> *Comment: “It seems like different sizes per group would be a more realistic expectation, and that perhaps this should be worked into the algorithm. Similarly, pooling only 2 groups together (from pre-specified positions) seems like it would be limiting as well. It also seems like the algorithm should account for being able to use a high-level feature from one layer as part of multiple groups in the future (i.e. reuse). Even if any of these options don’t make a difference, it would be good to check/evaluate.”
>
> Thank you for suggesting these ideas, we agree with you there are many dimensions that can be explored to extend the proposed method. However, in the current paper, we decided to present the main idea in a simple and clean way which would make the paper more practical.
> We think that extending the suggested idea to more complicated architectures and ideas such as sharing weights and groups would be a very interesting direction for any future study.
>
>
> **Minor:
> -------------------------------------------
> *Comment: “Equation 8 did not fully make sense to me.”
>
> To address your comment and to further clarify this for our readers, we added the following explanation to the revised paper (right after equation 8 in text):
> “, where the first term is the work required for the first network layer from $d$ to $km$ neurons, the second term is corresponding to a hidden layer of size $km$, and so forth. The last term is the complexity of the output layer similar to the case of GMLP.”
>
> -------------------------------------------
> *Comment: “Why were “random horizontal flips” used as preprocessing for the permutation-invariant CIFAR-10 dataset? This shouldn’t make a difference at all if the pixels become randomly shuffled anyway.”
>
> The random shuffle is happening in the data pipeline after the standard CIFAR-10 augmentation. Note that the random shuffle is not random across samples, it is merely a fixed random ordering used to remove pixel coordinates for each experiment.
> Also, we would like to emphasize that while we had the option to ignore any data augmentation for CIFAR-10, it is best for the comparison of results to use the established standard augmentation method for this dataset that is followed by the majority of other work in the literature.
>
> To address your concern and to prevent any confusion, we added the following clarification to the revised version (see the first paragraph of Section 4.1):
> “Note that the permutation is not changing across samples, it is merely a fixed random ordering used to remove pixel coordinates for each experiment.”
>
> Also, we changed “permutation invariant version” to “permuted version” in Table 2 and 3 captions.

---

> ### Author Response · Authors · 2019-11-14
> **To Reviewer #3 (1/2)**
>
>
> Thank you for reviewing the manuscript and helpful comments. Please find a point-to-point response to your comments in the following.
>
> -------------------------------------------
> *Comment: “The results do not show much improvement (i.e., < 0.3% improvement for 3 of the datasets, and < 1% for another one), aside from CIFAR-10. Considering that the premise of the paper is that MLP’s are not good enough when dealing with data in which the relationships between features are unknown, it seems like these are definitely not good datasets on which to demonstrate this notion of “there has been little progress in deep reinforcement learning for domains without a known structure between features.””
>
> We understand the reviewer’s concern that the improvements reported in Table 2 are often not larger than 1%. However, we would like to emphasize that for many of these tasks, which are in healthcare or UCI benchmarks, even a small percentage of improvement in accuracy is considered significant. For example, the original paper introduced self-normalizing neural networks (Klambauer et al. 2017) extensively used UCI benchmarks and often reported results that are only a fraction of percentage better than other competitors.
>
> Moreover, as demonstrated in Figure 2, the proposed architecture is much more efficient in terms of the number of parameters or model complexity at the prediction time. In other words, GMLP uses an order of magnitude less parameters and still achieves better results compared to the best MLP.
>
> Regarding the CIFAR-10 experiments, the use of a permuted version as a difficult task to challenge a non-image classifier is not new. For instance, very recent and relevant work by Aydore et al. 2019 used the same benchmark. Indeed, for CIFAR-10, we have more than 5% improvement over MLP which we believe is a new state-of-the-art for architectures not using pixel coordinates.
>
> -------------------------------------------
> *Comment: “The MNIST visualization of group-select felt informative, but the XOR example for grouping visualizations seemed too easy. It would’ve been good to see visualizations or intuitions regarding grouping for harder datasets, in order to be convinced of the need for more expressive feature representations than standard MLP’s.
>
> To address your comment, we added a new appendix to the revised paper: “Appendix D: Visual Analysis”. In this appendix, we visualize the selected features for 25 different groups from our CIFAR-10 architecture (see Figure 10) as well as a heatmap showing the overall frequency each pixel is being selected by the group-select layer (see Figure 11).
>
> Based on our initial investigation, it is difficult to objectively analyze the performance of the group-select layer on a large task such as CIFAR-10 which has 1536 groups of size 16 each. Note that often each group is selecting different locations from different channels which makes it challenging to interpret visually. However, one noticeable pattern is that features selected from a certain color channel usually appear in clusters resembling irregularly shaped patches.
>
> We understand the limitations of the current visualizations on small and simple tasks; however, we think these are still helpful conveying a general intuition on the operation of the suggested method.
>
> -------------------------------------------
> *Comment: I’m not an expert on causality, but it seems like citations from that area are required for this problem statement of dealing with features where the connections between them are unknown but potentially very important.”
>
> As suggested, we included a discussion of these methods in the paper and added proper citations.
>
> See the last sentence of Section 5 (discussion):
> “From another perspective, the idea of studying feature groups is closely related to causal models such as Bayesian networks and factor graphs (Darwiche, 2009; Neapolitan et al., 2004; Clifford, 1990). These methods are often impractical for large-scale problems because, without a prior over the causal graph, they require an architecture search of the NP-complete complexity or more.”

---

### Official Review · AnonReviewer4 · 2019-11-09
**Official Blind Review #4**

**Rating:** 3

**Review:**

Summary:

This paper gives a new MLP formulation and architecture that is ostensibly suited for data where the label's dependence on the input feature has form of sparsity. The paper reports the performance of this architecture on a few standard datasets and outperforms other baselines.

Review:

The main contribution arguably in the "Group select" matrix which selects features that each Group-FC focusses on. While it is an interesting idea, it is not hugely novel and requires a lot more demonstration. Without the Group-select the architecture in the paper can simply viewed as an MLP with a block diagonal sparsity structure enforced on the weights and max-out pooling as non-linearity.

Based on the above reasoning, I would rate the architectural/theoretical contribution as not significant. The empirical results also do not look particularly convincing, as there is just a few percentage improvement over MLP (except CIFAR10).

Questions for authors:

1. In the modified CIFAR10 dtaset, is the same permutation of pixels applied to all images or is it a different random permutation for every? If it is different, the entire rationale of Group-select breaks down, so I am assuming it is the same, but this must be clarified.

2. What were the groups selected in CIFAR10? Did the groups selected correspond to nearby pixels or some other meaningful way? A better understanding of why this improvement happened might help a lot.

3. In figure 5,  learning curves are plotted for different m. But what about k? Is it kept constant? If so what constant? Similarly for figure 6.

4. The visualisation for MNIST Group-select seems very limited compared to what is there in the architecture. I there any way a selected group can be shown and visualised in a meaningful way?

5. The synthetic example is interesting, but this can easily be extended to much larger scales. Do the results hold up in such cases too?

Typos/errors:

1. In equation 4, the i+k/2 part makes sense only for the l=0 layer.










**Experience Assessment:**

I have read many papers in this area.

**Review Assessment: Checking Correctness Of Derivations And Theory:**

N/A

**Review Assessment: Checking Correctness Of Experiments:**

I assessed the sensibility of the experiments.

**Review Assessment: Thoroughness In Paper Reading:**

I read the paper thoroughly.

---

> ### Author Response · Authors · 2019-11-14
> **To Reviewer #4**
>
> Thank you for reviewing the manuscript and helpful comments. Please find a point-to-point response to your comments in the following.
>
> -------------------------------------------
> *Comment: “1. In the modified CIFAR10 dtaset, is the same permutation of pixels applied to all images or is it a different random permutation for every? If it is different, the entire rationale of Group-select breaks down, so I am assuming it is the same, but this must be clarified.”
>
> The same permutation is used for all images. At the beginning of each experiment, we define a fixed permutation of features and use this to reorder all feature vectors. Here, the idea is to prevent a model to use pixel coordinates: it will make the task very difficult if not impossible for CNNs while it does not impose any challenge for an MLP.
>
> To address your concern and to prevent any confusion, we added the following clarification to the revised version (see the first paragraph of Section 4.1):
> “Note that the permutation is not changing across samples, it is merely a fixed random ordering used to remove pixel coordinates for each experiment.”
>
> Also, we changed “permutation invariant version” to “permuted version” in Table 2 and 3 captions.
>
> -------------------------------------------
> *Comment: “2. What were the groups selected in CIFAR10? Did the groups selected correspond to nearby pixels or some other meaningful way? A better understanding of why this improvement happened might help a lot.”
>
> To address your comment, we added a new appendix to the revised paper: “Appendix D: Visual Analysis”. In this appendix, we visualize the selected features for 25 different groups from our CIFAR-10 architecture (see Figure 10) as well as a heatmap showing the overall frequency of each pixel being selected by the group-select layer (see Figure 11).
>
> Based on our initial investigation, it is difficult to objectively analyze the performance of the group-select layer on a large task such as CIFAR-10 for which with have 1536 groups of size 16 each. Note that often each group is selecting different locations from different channels which makes it difficult to interpret visually. However, one noticeable pattern is that features selected from a certain color channel usually appear in clusters resembling irregularly shaped patches.
>
> -------------------------------------------
> *Comment: “3. In figure 5,  learning curves are plotted for different m. But what about k? Is it kept constant? If so what constant? Similarly for figure 6.”
>
> Thank you for pointing out this issue. We revised the paper (see captions in Figures 5 and 6) to report the exact m and k used in each case i.e., k=1536 for Figure 5, and m=16 for Figure 6.
>
> -------------------------------------------
> *Comment: “4. The visualisation for MNIST Group-select seems very limited compared to what is there in the architecture. I there any way a selected group can be shown and visualised in a meaningful way?”
>
> As we indicated in response to your earlier comment, in the revised version, we added a new appendix dedicated to the visual analysis on the group-select layer for our CIFAR-10 model.
>
> -------------------------------------------
> *Comment: “5. The synthetic example is interesting, but this can easily be extended to much larger scales. Do the results hold up in such cases too?”
>
> We agree with the reviewer that it would be interesting to extend this experiment; however, using a large scale and more complex dataset requires using a deeper GMLP with more groups and a larger group size. Based on our initial experiments, it will be challenging to make visual interpretations in such scenarios as groups start to learn with more redundancy, share information between the groups, and merge information deeper within the network.
>
> We understand the limitations of the current visualizations on small and simple tasks; however, we think these are still helpful conveying a general intuition on the operation of the suggested method.
>
> -------------------------------------------
> *Comment: “Typos/errors:
>
> 1. In equation 4, the i+k/2 part makes sense only for the l=0 layer.”
>
> Thank you for pointing out this error. We revised equation (4) to fix this issue:
>
> $$
>     \bm{z}_{i}^{l+1} = pool(\bm{z}_{i}^{l},\bm{z}_{i+k/{2^{l+1}}}^{l})
> $$

---

### Decision · Program_Chairs · 2019-12-19

**Decision:**

Reject

**Comment:**

The authors propose Group Connected Multilayer Perceptron Networks which allow expressive feature combinations to learn meaningful deep representations. They experiment with different datasets and show that the proposed method gives improved performance.

The authors have done a commendable job of replying to the queries of the reviewers and addresses many of their concerns. However, the main concern still remains: The improvements are not very significant on most datasets except the MNIST dataset. I understand the author's argument that other papers have also reported small improvements on these datasets and hence it is ok to report small improvements. However, the reviewers and the AC did not find this argument very convincing. Given that this is not a theoretical paper and that the novelty is not very high (as pointed out by R1) strong empirical results are accepted.  Hence, at this point, I recommend that the paper cannot be accepted.